# Functional Imaging in the Evaluation of Treatment Response in Multiple Myeloma: The Role of PET-CT and MRI

**DOI:** 10.3390/jpm12111885

**Published:** 2022-11-10

**Authors:** Adele Santoni, Martina Simoncelli, Marta Franceschini, Sara Ciofini, Sara Fredducci, Federico Caroni, Vincenzo Sammartano, Monica Bocchia, Alessandro Gozzetti

**Affiliations:** Hematology, Azienda Ospedaliera Universitaria Senese, University of Siena, 53100 Siena, Italy

**Keywords:** multiple myeloma, PET/CT, MRI, treatment response, minimal residual disease, MRD

## Abstract

Bone disease is among the defining characteristics of symptomatic Multiple Myeloma (MM). Imaging techniques such as fluorodeoxyglucose positron emission tomography–computed tomography (FDG PET/CT) and magnetic resonance imaging (MRI) can identify plasma cell proliferation and quantify disease activity. This function renders these imaging tools as suitable not only for diagnosis, but also for the assessment of bone disease after treatment of MM patients. The aim of this article is to review FDG PET/CT and MRI and their applications, with a focus on their role in treatment response evaluation. MRI emerges as the technique with the highest sensitivity in lesions’ detection and PET/CT as the technique with a major impact on prognosis. Their comparison yields different results concerning the best tool to evaluate treatment response. The inhomogeneity of the data suggests the need to address limitations related to these tools with the employment of new techniques and the potential for a complementary use of both PET/CT and MRI to refine the sensitivity and achieve the standards for minimal residual disease (MRD) evaluation.

## 1. Introduction

Multiple Myeloma (MM) is a plasma-cell malignancy in which a single clone of plasma cells proliferates and produces a monoclonal protein (MC). Myeloma clonal plasma cells accumulate in the bone marrow (BM), resulting in diffuse skeletal involvement, hypercalcemia, anemia and extramedullary localizations [1,2,3,4]. In addition, MC and free-light chains are nephrotoxic and may result in renal failure. Many advances have been made in MM treatment even in aggressive forms of disease [5,6,7,8,9]. Early diagnosis of organ damage has recently been improved. Symptomatic MM is described by the typical CRAB features (hypercalcemia, renal failure, anemia, bone disease) where bone disease is now referred to as >1 osteolytic lesion (≥5 mm) on skeletal radiography, computed tomography (CT) or positron emission tomography with computed tomography (PET/CT) [10,11] as one of the myeloma defining events (MDE) defined by the International Myeloma Working Group (IMWG) which can identify active disease and the need to treat patients [12]. In addition, creatinine 2 mg/dL could be a MDE although renal involvement can be seen at the MGRS stage (monoclonal gammopathy of renal significance (MGRS)) [13]. Bone disease is indeed a mainstay of active MM; hence, it is identified in 70% of patients and invariably indicates a need to start therapy [14]. The established imaging modalities to manage MM patients are CT, Magnetic Resonance Imaging (MRI) and PET/CT, whereas there is consensual agreement on their replacement of the old X-ray skeletal survey because of its significantly lower sensitivity for bone lesions’ detection. Recently, MRI and PET/CT have acquired a major role with respect to low dose CT. These imaging modalities can functionally quantify bone disease before bone damage, and lessons can be obtained from the treatment of lymphomas [15]. MM is characterized by the focal and patchy distribution of plasma cells in the BM, the focal lesions (FLs) identified with MRI and PET/CT are focal accumulations of plasma cells and they are different from the lytic lesions detected with low dose CT, where bone destruction has already occurred [16]. FLs are defined as focal bone uptake on two consecutive PET slices without evident changes on CT for PET/CT and a low T1-weighted signal and a high T2w-STIR signal on MRI [17]. Functional imaging modalities can even describe other disease patterns; specifically, MRI is able to accurately identify diffuse BM infiltration thus identifying both diffuse diseases, FLs and their possible combination [18]. The key role of both imaging modalities in the identification of myeloma patients needing treatment has the potential to be extended to comprehensive MM management. The ability to quantify bone disease in terms of functional activity, renders MRI and PET/CT as potential tools to evaluate treatment response. PET/CT has been identified by the IMWG as the gold standard for evaluation of minimal residual disease (MRD) after therapy.

## 2. PET/CT: Methods and Role in Response to Therapy in MM

^18^F-Fluorodeoxyglucose (FDG)-PET/CT is considered the best functional imaging technique in defining the metabolic activity of bone lesions, plasmacytomas and extra-medullary disease (EMD) caused by MM. ^18^F-FDG is currently the most used tracer for MM and it is a radiolabeled glucose analogue in which the C-2 hydroxyl group is replaced by the positron-emitting fluorine-18 atom (^18^F) [19]. FDG is taken up by high-glucose-using cells through the glucose transporters GLUT1 and GLUT3 and it is phosphorylated by a hexokinase to ^18^F-FDG-6P, that is stored in cells as it cannot be further metabolized. After intravenous administration, ^18^F-FDG reaches an equilibrium state in 60 min and it starts to decay in 80–90 min, allowing an accurate assessment of glucose metabolic activity. MM plasma cells usually overexpress hexokinase-2, displaying a higher glycolytic activity than surrounding cells [17]. PET positivity is defined as the presence of areas of ^18^F-FDG uptake higher than the liver uptake, that is taken as a background with a Deauville score of 4. Consequently, PET positive lesions have a Deauville Score of 4 or 5 [20]. For a better localization of metabolically active FLs, PET is usually combined with low dose CT (120 kV, 80 mA). Mostly the field of view (FOV) includes the region spanning from the skull to the femur, encompassing the upper limbs, and there are only a few centers using whole body (WB) PET/CT [21]. ^18^F-FDG-PET/CT is used to clarify a dubious diagnosis between MM and smoldering multiple myeloma (SMM), when low-dose WB-CT and WB-MRI are inconclusive but there is a strong suspicion of active MM, because of the high cost of this imaging technique. ^18^F-FDG-PET/CT is also used for the diagnosis of suspected extramedullary solitary plasmacytoma [6] and in the prognostic assessment of newly diagnosed MM (NDMM): more than three FLs [22], EMD and a maximum standardized uptake value (SUVmax) greater than 4.2 are related to a poor outcome [21]. These three factors are independently associated with progression-free survival (PFS). High FDG-avidity (SUV > 4.2) and the presence of EMD are also correlated with shorter overall survival (OS) [23]. The main implication of ^18^F-FDG-PET/CT is in the assessment of treatment response. After the first line therapy, patients undergo a repeat bone marrow (BM) aspiration, for the evaluation of MRD, and ^18^F-FDG-PET/CT is the only imaging technique able to distinguish between active and inactive FLs, with high prognostic value: ^18^F-FDG uptake in bone lesions has higher sensitivity in detecting tumor residual disease than immunofixation electrophoresis [24] and appears to be closely related with PFS and OS. In the standardization of the ^18^F-FDG-PET/CT Deauville criteria for MM, Zamagni et al. defined the complete metabolic response as the ^18^F-FDG uptake minus the liver activity in BM sites and FLs previously involved (including extramedullary and para-medullary disease (DS score 1–3)) and partial metabolic response as the decrease in number and/or activity of BM/FLs present at baseline, but persistence of lesion(s) with an uptake greater than the liver activity (DS score 4 or 5). The metabolic disease is defined as stable if there are no significant changes in BM/FLs and progressive when new FLs appear compared with baseline [25].

### 2.1. New Tracers

The sensitivity of ^18^F-FDG-PET/CT in bone lesions’ detection at diagnosis ranges from 59% to 100%, whereas the specificity ranges from 75% to 82% [21]. False positive can be obtained in the case of other malignancies, inflammation, infections, fractures, post-surgical areas (including BM biopsy), recent chemotherapy infusion and use of growth factors, therefore PET/CT should be performed at least one month after the use of these agents [25]. False negative can be obtained in the case of administration of high-dose steroids, hyperglycemia [21] and in cases of a lack of hexokinase that can occur in 10–15% of MM patients. To overcome this limitation, new tracers have been evaluated for patients with MM, specific for plasma cells or proliferating cells or conjugated with anti-CD38 Daratumumab (immune PET tracers). Lipid tracers, such as choline and acetate, were first investigated. Choline is an indicator of the synthesis of plasma membrane, and it can be labelled with ^11^C and ^18^F, becoming more sensitive than ^18^F-FDG [26,27]. Indeed, acetate is converted in acetyl-CoA in Krebs’ cycle, producing energy for cells and it is also considered more sensitive than ^18^F-FDG [28,29]. Amino acid tracers are used by plasma cells in the synthesis of new proteins, such as immunoglobulins, and they can be more specific biomarkers of active MM. Among them, ^11^C-methionine is more sensitive than ^18^F-FDG and ^11^C-choline [30], even if it has a short half-life (20 min) and an on-site cyclotron is necessary to produce it [31]. Stokke et al. demonstrated that ^18^F-fluciclovine, a leucine analogous, is also a promising tracer for MM, reaching a similar uptake pattern and similar sensitivity, with a half-life of 110 min [32]. ^18^F-fluoro-ethyl-tyrosine could be used as a tracer, but it is less sensitive than ^11^C-metionine and ^18^F-FDG, having minimal transportation into plasma cells [33]. Nucleoside tracers are related to the rate of DNA synthesis, reflecting highly proliferating cells. ^11^C-thiothymidine sensitivity is similar to the ^11^C-methionine one and it is better than ^18^F-FDG sensitivity, especially during the early stage of the disease [34]. ^18^F-fluorothymidine was investigated as a new tracer, but it is not considered a useful biomarker for MM [35]. ^18^F-fluoride reflects the early phase of bone calcification, but results concerning the diagnostic assessment in MM are divergent [36]. ^89^Zr-bevacizumab, a radiolabeled antibody directed to the VEGF receptor, is uniformly expressed on plasma cells and it could be useful for the detection of MM [37]. In the end, there are also tracers targeting molecules expressed on cells surface such as CD38, expressed on plasma cells, and CXCR4, a chemokine expressed on hematopoietic stem cells. The activation of CXCR4/stromal-derived factor 1 axis correlates with bone activation, playing an important role in MM. Pentixafor radiolabeled with gallium-68 is used to target CXCR4. Its sensitivity compared to ^18^F-FDG is not clear, but a positive ^68^Ga-pentixafor has a negative prognostic significance, with a poorer OS [38]. CD38 is highly expressed on all MM cells and anti-CD38 monoclonal antibody (Daratumumab) can be labelled with Zirconium-89 or Copper-64 to target them, but data are not yet available from recent clinical trials.

### 2.2. PET/CT and MRD in MM

Due to the patchy dissemination of the disease, MRD evaluation in the BM could sometimes lead to false negative results in the presence of minimal disease after therapy [39]. In the absence of recognized new tools for blood MRD evaluation, PET is a valid technique able to identify focal active metabolic lesions that can be reservoirs for MM relapse.

MRD negative is a deeper level of response than a complete response, that requires the absence of phenotypically abnormal clonal plasma cells from BM aspirates, detected by next-generation flow cytometry (NGF) or next-generation sequencing (NGS). However, MM is a heterogenous disease with patchy infiltration and EMD is not uncommon. The association between BM analysis and ^18^F-FDG-PET/CT leads to a more accurate assessment of response after a treatment line with the highest prognostic value, well related to PFS and OS. In fact, although MRD-negativity is associated with improved outcomes, in MRD-negative patients relapse still occurs, potentially due to the presence of focal bone disease that could be detected with ^18^F-FDG-PET/CT. The IFM 2009 trial showed patients who were double negative (with no residual disease assessed by NGF and negative PET/CT) achieving better PFS than patients who were not double negative [38]. Alonso et al. selected 103 NDMM patients who received their first-line therapy, underwent BM MRD assessment with NGF and ^18^F-FDG-PET/CT evaluation at diagnosis and one month after the end of the treatment. It was observed that patients MRD-/PET- had the best 4-years OS (94.2%) and PFS (92 months), patients MRD+/PET− had a not significant difference in 4-years OS (100%) but a shorter PFS (45 months) and PET+ patients had the worst 4-years OS (73.8%) and PFS (28 months) [40]. This study shows that ^18^F-FDG-PET/CT positivity after first line treatment is the most affecting factor on PFS and OS.

### 2.3. Clinical Studies

Numerous studies evaluated the accuracy of ^18^F-FDG-PET/CT in the therapy response assessment, even if they are not exactly concordant at the time of evaluation and in the interpretation of results. The main disadvantages of the ^18^F-FDG-PET/CT, beyond the high cost and the limited availability, are the lack of standardization criteria and of interobserver reproducibility in the interpretation of results. More than 30 clinical studies on the prognostic role of PET/CT in the evaluation of treatment response were found. In Table 1, we report on only a few studies with more than one hundred patients enrolled; despite the possible difference in obtaining data from clinical studies, these and all the other studies that are not reported for a rare number of patients enrolled agree with the prognostic significance of PET/CT in the evaluation of response to therapy. 

## 3. MRI: Methods and Role in Response to Therapy in MM

After therapy, the sensitivity of BM samples is reduced because marrow involvement in MM could be patchy, while MRI shows a better assessment for detecting diffuse marrow involvement, correlating with response to therapy [50]. WB-MRI assesses water and fat in tissues and it can detect BM abnormalities before significant bone destruction. It is used to detect FLs as MDE, according to the European Society for Medical Oncology (ESMO) guidelines, [51] and it is also considered the first-line imaging choice for solitary plasmacytoma confined to bone.

WB-MRI can be divided into anatomical MRI sequences, fat and water fraction sequences, diffusion-weighted imaging (DWI) and dynamic contrast-enhanced (DCE) MRI. A typical MRI pattern in MM is characterized by hypo-intensity and high contrast-enhancement in T1-weighted, hyper-intensity in T2-weighted and signal loss in the opposite phases [52]. MRI can identify five BM infiltration patterns in MM: normal BM; salt and pepper; diffuse; and combined diffuse and focal pattern [53].

WB-MRI typical protocols are various. Anatomic sequences such as T1- and T2-weighted imaging could be used for identification and measurement of focal marrow replacing lesions [54]. The Dixon technique provides axial anatomical details and it is based on the differences in resonance frequency of fat and water hydrogen nuclei [55], while axial diffusion is the most sensitive and it forms an apparent diffusion coefficient (ADC) map [45]. Finally, they include the total body (vertex to toes) in MRI [56]. Despite the improved diagnostic performance of the MRI, the conventional sequences do not provide adequate functional information that is necessary for the assessment of treatment response, because they consider only the anatomic and the morphologic assessment. For this reason, functional MRI sequences have been developed for treatment response assessment of patients with MM. Among these, DCE imaging evaluates the skeleton using a time series of images after intravenous injection of gadolinium contrast and it can provide functional information on BM vascularization that changes in different disease stages, giving an analysis of dynamic images in a native review method, a qualitative, a semi-quantitative or a quantitative evaluation. The native review method analyzes images at different time points after gadolinium administration to find differences in contrast enhancement [57]. The other methods require post-processing images: the region of interest (ROI) is manually placed in disease tissues, and they acquire time-intensity curves that reflect the passage of gadolinium from the intravascular to the interstitial space. These curves could be evaluated both qualitatively, scoring the curve pattern, and semi-quantitatively, calculating the descriptive parameters [58]. In MM, DCE-MRI shows an increased microcirculation which correlates with disease progression, and it gives additional information to the anatomical MRI [59]. However, DCE-MRI has some limitations as BM vascularization depends on various factors such as age, sex, BMI and bone density, that influence the distribution pattern of yellow and red BM. The latter is characterized by numerous vessels and consequently by an increased enhancement; it is typical of hematopoietic and pathologically infiltrated BM and it decreases with increasing age [60]. Therefore, it is necessary to adapt normal values for age, sex and anatomic location to avoid misinterpretation and some studies are necessary to have a standardization. Furthermore DCE-MRI depends on the observer variability, and it is difficult to objectify correlation with clinical endpoints [58]. Consequently, DCE-MRI is not currently indicated in the IMWG guidelines on imaging in plasma cell disorders. Another method is DWI, that is used to assess BM diseases because it can assess tissue cell density, detecting free water molecule movements depicted on ADC [60]. The power of the diffusion depends on the diffusion sensitizing gradient, expressed as b-value: higher b-values identify FLs better than the lower b-values [61]. Marrow infiltration increases cellularity, resulting in limited movements and decreased Brownian motion of free water molecules, increased signal on DWI sequence and decreased ADC values [62]. In this way, DWI can provide a visual contrast between normal and infiltrated marrow, performing a qualitative evaluation, but it can also provide a quantitative assessment calculating ADC values and ADC maps [63]. DWI makes a functional evaluation of bone lesions of MM, providing a measure of tumor cellularity and additional information on microenvironmental changes, but it reveals nothing about the cell viability. Some studies indicate that changes in the ADC value are associated with response to treatment, before the size reduction and the number of lesions [64]. In particular, lower ADC before treatment is potentially associated with a better treatment response [65]. The DWI role in treatment-response assessment of MM is evaluated by a recent meta-analysis: they found that DWI has a sensitivity of 78% and a specificity of 73% in distinguishing responders and non-responders [50]. Despite that, this technique could present artifacts and it could give possible false positives, because sometimes lesions undergo necrotic changes and they are still visible, even if the cells are not vital. In addition, ADC in DWI is influenced by diffusion but also by perfusion and it is a disadvantage. For all these reasons, DWI should be combined with other sequences in WB-MRI to obtain a better assessment.

In conclusion, functional MRI sequences help to better characterize FLs and they have a prognostic value in MM; in particular, DWI is a promising tool for response assessment after treatment in this setting.

### 3.1. MRI and MRD in MM

NGF with functional imaging has been used to define responses in MM [66], though the role of MRI to complement MRD needs confirmation. The use of MRI for the assessment of MRD is not clear because there are no data for a comparison with FDG-PET/CT in evaluating MRD after therapy. Moving forward, it is crucial to standardize guidelines about the use of imaging techniques and the time point to be assessed. It is necessary to establish the role of MRI to the definition of MRD with new comparative studies.

### 3.2. Clinical Studies

Although IMWG guidelines indicate the use of DWI-MRI to evaluate plasma cell disorders, the use of MRI to assess response to treatment is a matter of debate and it is investigated in several clinical studies, in small series of patients. In recent years, some studies have researched the role of DWI in treatment response assessment in patients with MM (Table 2). They have proven that DWI is more sensitive than the conventional MRI and it gives a quantitative and non-invasive evaluation of BM after therapy, distinguishing responders from non-responders. Some of these studies also hypothesize that WB-DWI could be equivalent or superior to PET/CT assessment of the response and the MRD. To shed light on the role of WB-MRI, a multidisciplinary, international, and expert panel of radiologists, medical physicists and hematologists developed the Myeloma Response Assessment and Diagnosis System (MY-RADS) imaging recommendations [67].

Despite these current data, a better definition of the DWI-WB-MRI role in treatment assessment is necessary. Nowadays some prospective studies are ongoing to determine the sensitivity, specificity and the accuracy of WB-MRI. Additionally, some investigators are studying in MM patients the Dual-Echo T2 weighted acquisition for Enhanced Conspicuity of Tumors, an alternative WB-MRI technique that improves tumor visualization overcoming the compromised image quality of DWI-MRI (NCT04493411).

## 4. Comparison of PET/CT and Functional MRI in Response Evaluation in MM

Evidence shows that both MRI and PET/CT have significant potential in evaluating treatment response as separate tools. Specifically, data from PET/CT studies point out its prognostic and predictive value (Table 1) whereas MRI emerges as particularly sensitive technique to detect myeloma FLs and background marrow infiltration, with the addition of DWI bringing further improvement in sensitivity and potential for differentiation between active and treated disease in cases of disease relapse. 

Several studies have also investigated which modality performs better in the setting of treatment-response evaluation. A recent comparative meta-analysis of Yokoyama et al. has shown the major impact of PET/CT rather than MRI in response to therapy (sensitivity 80% vs. 20%, specificity 58% vs. 83%) [80]. Other studies reach the same conclusions even with different data (same sensitivity 75%, specificity 86% vs. 43%) [81]. The prognostic value of PET/CT and WB-MRI has also been investigated for response evaluation at different time points of treatment, finding only PET/CT significantly able to predict response to autologous stem cell transplantation (ASCT) in the post-induction phase [32,34,71] and OS and PFS in the post-transplant evaluation [82]. The glucose metabolism exploited by FDG-PET/CT rapidly follows the dynamic change of FLs during treatment, whereas the persistence of non-viable lesions on MRI images may explain its reduced specificity and the lack of prognostic value [83,84]. Sometimes active lesions may be misinterpreted as scar tissue, affecting sensitivity [80]. Implementation with DWI technique increases the specificity of MRI sections [81], though it seems to not gain advantage over PET/CT in predictive value. On the other hand, another recent meta-analysis of data from 12 comparative studies assessing the accuracy of WB-MRI and FDG-PET/CT, identify the first as the most sensitive technique (90% vs. 66%) for determining response through earlier detection of post-treatment recurrence, though the finding was not significant. Similar evidence is reported in a retrospective study comparing PET/CT and MRI in different treatment phases [83]. These results could also be biased from the delayed healing of pre-treatment myeloma lesions which can still be detected from MRI. The bias has been addressed by the meta-analysis investigator Rama et al. by considering five studies comparing PET/CT with DWI-WB-MRI. The use of DWI though displayed the same specificity of WB-MRI (57% vs. 56%), not overcoming the limitation. 

Indeed, the studies reviewed in these articles display controversial data for techniques comparison underlining the need to understand and address MRI and PET/CT limitations (Table 3).

### Comparison of PET/CT and Functional MRI for MRD in MM

Concerning comparison of PET/CT and functional MRI for MRD assessment, clinical studies are few. According to IMWG, PET/CT represents the best tool for MRD evaluation as it has been shown to predict survival after therapy. However, most clinical trials using PET/CT as MRD tool are still ongoing and preliminary data do not show significant concordance between imaging and BM assessment [85]. A recent study from Rashe et al. evaluating the combination of functional imaging and flow cytometry for MRD evaluation on a cohort of patients undergoing first or successive lines of therapy, proposed DWI-MRI as complementary imaging tool to PET/CT, as the PFS of patients with FLs detected with DWI-MRI or PET/CT were not statistically significant (3.4 years vs. 3 years). Furthermore, the study identified that the combination of both approaches yielded the highest rate of FLs detection in patients achieving CR [86].

Even though recommendations for the use of different techniques may vary depending on available local resources, we personally use PET/CT at diagnosis and at the end of a fixed treatment therapy (3 months after ASCT) and then every 6 months for 2 years. We integrate radiological study with MRI when PET is negative for disease and when local vertebroplasty or radiotherapy is needed. 

## 5. Conclusions

In conclusion, in agreement with current data coming from comparative studies, PET/CT is probably the best choice for the evaluation of treatment response, bringing also prognostic value. A condition where DWI-WB-MRI sensitivity overcomes FDG-PET is disease with low expression of hexokinase-2, involved in the glycolytic pathway, as ^18^F-FDG-PET/CT yields false negative results losing sensitivity [87]. In addition, emerging studies on new tracers-based PET/CT (chemokine receptor 4–targeted PET/CT with gallium 68–Pentixafor or CD38-targeted PET/CT with zirconium 89-DFO-daratumumab) display potential for overcoming this limitation of disease conditions [88,89]. To gain both sensitivity and specificity in lesions’ detection, exploiting the complementarity of PET and MRI as unique modalities to evaluate treatment response is something to examine in future studies [90]. Furthermore, functional images and particularly PET/CT is the indicated tool together with NGF for the evaluation of medullary and extramedullary MRD in MM [85]. Combination of both PET and MRI may provide a deepening of sensitivity which could contribute to better outline functional imaging as a tool for MRD evaluation.

## Figures and Tables

**Table 1 jpm-12-01885-t001:** Main studies assessing the performance of ^18^F-FDG-PET/CT for treatment response in MM patients.

Study, Year, Journal	Type	Patients	Aim	Results
Bartel et al., 2009, *Blood* [41]	Prospective	239	To examine the prognostic implications of ^18^F-FDG-PET/CT ^1^ and MRI ^2^ in patients (pts) with NDMM ^3^	30 months OS ^4^: 92% if complete suppression in FL ^5^ and EMD ^6^ after CTx ^7^ 71% if not complete suppression after CTx (*p* 0.0002) 30 months PFS ^8^: 89% if complete suppression after CTx 63% if not complete suppression after CTx (*p* 0.0003)
Zamagni et al., 2011, *Blood* [23]	Prospective	192	To determine prognostic implications of ^18^F-FDG-PET/CT at diagnosis and in the evaluation of treatment response	4 years PFS: PET−: 47% PET+: 32% (*p* 0.02) 4 years OS: PET−: 79% PET+: 66% (*p* 0.02)
Nanni et al., 2013, *Clinical Nuclear Medicine* [42]	Prospective	107	To analyze the prognostic value of ^18^F-FDG-PET/CT after therapy in patients with MM ^9^	TTR ^10^ in relapsed pts (44%): PET− after ASCT ^11^: 27.6 mo. PET+ after ASCT: 18 mo. (*p* 0.05)
Usmani et al., 2013, *Blood* [43]	Prospective	302	To investigate the survival implications of the day 7 PET scanning of patients treated with total therapy 3A clinical trial and Total Therapy 3B protocol	3 years PFS: 0 PET-FL: 84% 1–3 PET-FL: 78% >3 PET-FL: 56% (*p* 0.0003) 3 years OS: 0 PET-FL: 87% 1–3 PET-FL: 82% >3 PET-FL: 63% (*p* < 0.0001)
Zamagni et al., 2015, *Clinical Cancer Research* [44]	Retrospective	282	To evaluate the role of ^18^F-FDG-PET/CT in a cohort of symptomatic MM patients treated up-front	PFS: PET−: 52 months PET+: 38 months (*p* 0.0319) 5 years OS: PET−: 90% PET+: 71% (*p* 0.0014)
Moreau et al., 2017, *Journal of Clinical Oncology* [45]	Prospective	134	To assess the prognostic impact of MRI and PET/CT regarding PFS and OS	30 months PFS: PET−: 78.7% PET+: 56.8% (*p* 0.08) 2 years OS: PET−: 94.2% PET+: 72.9% (*p* < 0.001)
Zamagni et al., 2018, *Blood* [46]	Prospective	236	To standardize PET/CT evaluation by centralized imaging and revision, to define criteria for PET negativity after therapy (MRD ^12^ definition), evaluating ^18^F-FDG-PET/CT at diagnosis and prior to maintenance therapy in a sub-group of patients with NDMM	PFS: FL < 3: 40 months FL > 3: 26 months (*p* 0.0019) BMS ^13^ < 3: 39.8 months BMS > 3: 26.6 months (*p* 0.024) 63 months OS: FLs < 3: 73% FLs > 3: 63.6% (*p* 0.028) BMS < 3: 75.5% BMS < 3: 49.7% (*p* 0.002)
Zamagni et al., 2021, *Journal of Clinical Oncology* [47]	Retrospective	228	To standardize ^18^F-FDG-PET/CT according to Deauville criteria	PFS: FS < 4: 40 months FS > 4: 26.6 months (*p* 0.0307) BMS < 4: 44.9 months BMS > 4: 26.6 months (*p* 0.028) 60 months OS: FS < 4: 77.7% FS > 4: 64.1% (*p* 0.0276) BMS < 4: 76.7% BMS > 4: 52.1% (*p* 0.029)
Kaddoura et al., 2021, *Blood Advances* [48]	Retrospective	229	To determine prognostic impact of post-transplant, day 100 PET/CT scan	TTP ^14^: PET−: 24 months PET+: 12.4 months (*p* < 0.0001) OS: PET−: 100 months PET+: 47.2 months (*p* < 0.0001)
Charalampos et al., 2022, *Blood* [49]	Retrospective	195	Prognostic significance of PET/CT at 6 months following induction therapy	TTNT ^15^: CR ^16^, PET−: 58.9 mo. CR, PET+: 39.2 mo. (*p* 0.27) >VGPR ^17^, PET−: 46.9 mo. >VGPR; PET+: 26.9 mo. (*p* 0.02) <VGPR, PET−: 55.2 mo. <VGPR, PET+: 50.4 mo. (*p* 0.0058) OS: CR, PET−: unreached CR, PET+: 72 mo. (*p* 0.01) >VGPR, PET−: unreached >VGPR; PET+: unreached (*p* 0.00051) <VGPR, PET−: 112.7 mo. <VGPR, PET+: 9.5 mo. (*p* 0.032)

^1 18^F-FDG-PET/CT: ^18^-fluorodeoxyglucose positron emission tomography/computed tomography; ^2^ MRI: magnetic resonance imaging; ^3^ NDMM: newly diagnosed multiple myeloma; ^4^ OS: overall survival; ^5^ FL: focal lesion; ^6^ EMD: extra-medullary disease; ^7^ CTx: chemotherapy; ^8^ PFS: progression-free survival; ^9^ MM: multiple myeloma; ^10^ TTR: time to relapse; ^11^ASCT: autologous stem cell transplantation; ^12^ MRD: minimal residual disease; ^13^ BMS: bone marrow Deauville score; ^14^ TTP: time to progression; ^15^ TTNT: time to next treatment; ^16^ CR: complete response; ^17^ VGPR: very good partial response.

**Table 2 jpm-12-01885-t002:** Main studies assessing the performance of DWI-MRI for treatment response in MM patients.

Study, Year, Journal	Type	Patients	Aim	Results
Fenchel et al., 2010, *Acad Radiol.* [68]	prospective	10	To determine response to therapy using non-contrast perfusion MRI ^1^ and DWI WB-MRI ^2^	Mean diffusion increased after therapy Baseline 0.68 ± 0.19 × 10^3^ s/mm^2^ After treatment 0.96 ± 0.40 × 10^3^ s/mm^2^
Horger et al., 2011, *Am J Roentgenol.* [69]	prospective	12	To assess the feasibility of DWI WB-MRI for the evaluation of response to treatment in MM ^3^	ADC ^4^ value changes after treatment Responders 63.92% Non-responders 7.82%
Messiou et al., 2012, *Br J Radiol* [70]	prospective	20	To determine the response to treatment in MM patients	ADC value Active disease 0.76 ± 0.25 × 10^3^ s/mm^2^ After treatment 0.60 ± 0.46 × 10^3^ s/mm^2^
Giles et al., 2014, *Radiology* [71]	prospective	26	To determine the feasibility of DWI-MRI for assessment of treatment response in MM	ADC value changes after treatment Responders 19.80% Non-responders 3.20%
Bonaffini et al., 2015, *Acad Radiol* [72]	prospective	14	To determine the value of DWI-MRI in the assessment of response to chemotherapy in patients with MM	ADC value changes after treatment Responders 66% Non-responders 15%
Latifoltojar et al., 2017, *BJHaem* [73]	prospective	25	To explore and compare FLs ^5^ measures and DWI WB-MRI before and after chemotherapy in NDMM ^6^	ADC value changes after treatment Baseline 0.75 × 10^3^ s/mm^2^ After treatment 1.34 × 10^3^ s/mm^2^
Dutoit et al., 2016, *Eur J Radiol.* [58]	prospective	68	To evaluate the value of DCE-MRI ^7^ and DWI in MM patients after treatment	Agreement between IMWG ^8^ and MRI response criteria Kendall’s coefficient = 0.761
Wang et al., 2017, *CJH* [74]	prospective	8	To explore the practical value of WB-DWI in the diagnosis and monitoring of NDMM patients	ADC value changes after treatment Baseline 0.984 × 10^3^ s/mm^2^ After treatment 1.142 × 10^3^ s/mm^2^
Latifoltojar, 2017, *Eur Radiol.* [75]	prospective	21	To evaluate association between DWI WB-MRI and treatment response in MM	ADC value changes after treatment Baseline 0.804 × 10^3^ s/mm^2^ After treatment 1.180 × 10^3^ s/mm^2^
Lacognata et al., 2017, *Clin Radiol* [76]	prospective	18	To evaluate the modification of DWI WB-MRI after induction chemotherapy in MM patients and to correlate with patients response to therapy	ADC value changes after treatment Responders 32% Non-responders 6%
Wu et al., 2018, *Acad Radiol.* [77]	prospective	17	To assess the diagnostic accuracy of WB-DWI MRI in evaluation of response to induction chemotherapy in MM	ADC value changes after treatment Responders 36.79% Non-responders 11.50%
Park et al., 2020, *Cancer Imaging* [78]	retrospective	75	To evaluate the role of WB-DWI in the response assessment	Agreement between clinical and imaging response K = 0.69 for MDA-DWI ^9^ criteria
Takasu et al., 2020, *PLoS One* [79]	prospective	50	To compare remission status at the end of chemotherapy using WB-DWI in MM To assess the predictive value of MRI	ADC value changes after treatment Responders 25.50% Non-responders 1.46%
Costachescu et al., 2021, *Exp Ther Med.* [64]	retrospective	32	To evaluate DWI-WB MRI as possible prognostic factor in patients with MM	ADC values are inversely correlated with OS ^10^ r −0.641, *p* < 0.001

^1^ MRI: magnetic resonance imaging; ^2^ DWI WB-MRI: diffusion-weighted imaging whole body-MRI; ^3^ MM: multiple myeloma; ^4^ ADC: apparent diffusion coefficient; ^5^ FLs: focal lesions; ^6^ NDMM: newly-diagnosed multiple myeloma; ^7^ DCE-MRI: dynamic contrast-enhanced-MRI; ^8^ IMWG: International Myeloma Working Group; ^9^ MDA-DWI: MD Anderson-DWI; ^10^ OS: overall survival.

**Table 3 jpm-12-01885-t003:** Summary of advantages and limitations of PET/CT and functional MRI.

Modality	Advantages	Limitations
**PET/CT**	Concurrent morphologic and functional assessment	False negative in low hexokinase expression MM
	Quantification of disease metabolic activity	False positive in inflammatory setting
	Post-therapeutic prognostic significance	Expensive
		Lack of standardization
**Functional MRI**	Gold standard for diffuse BM infiltration	False positive for persistence of not active lesions
	Apparent highest sensitivity in lesions detection	Lack of standardization Need for integration with other MRI sequences

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
