# Peer review of "Functional Imaging in the Evaluation of Treatment Response in Multiple Myeloma: The Role of PET-CT and MRI"

_jpm, 2022, doi:10.3390/jpm12111885_

Round 1

Reviewer 1 Report

Santoni et al present an outstanding review manuscript on the role of PET-CT and MRI in the diagnosis and treatment response evaluation in multiple myeloma. They give a broad overview of the topic with with enough detail to be useful to radiologists, nuclear medicine imagers, and oncologists. 

Specific critiques

Practicing physicians can base their radiology testing on the data presented in the paper. It might be especially useful in the closing paragraphs to describe the exact imaging protocols the authors or their collaborators use. For example, MRI at question of diagnosis, PET/CT prior to start of therapy, repeat PET/CT after 3 cycles of induction therapy.

Along those lines, in the Conclusions, the authors bring up low expressing hexokinse-2 tumors as an explanation for false-negative PET/CT in myeloma, about 10%. How to use this knowledge prospectively? Perhaps a algorithm diagram - if pre-treatment PET/CT is negative, no use in repeating at 3 months, perhaps use MRI, BMBx in that case?

Line 339:  "yields false negative results losing specificity." Should be losing sensitivity, no?

The English is overall outstanding but some minor edits:

Line 38: "Many progresses have been made in MM" should be "Many advances"

Lines 257-257: "NGF with functional imaging to define response in MM, though the role of MRI to complement MRD needs confirmation." Sentence fragment. Perhaps, "NGF with functional imaging has been used to define.."

Line 259: "Nowadays it is crucial to have a standardization of guidelines" is awkward. Maybe, "Moving forward, it is crucial to standardize guidelines?"

Author Response

Reviewer1.

Santoni et al present an outstanding review manuscript on the role of PET-CT and MRI in the diagnosis and treatment response evaluation in multiple myeloma. They give a broad overview of the topic with with enough detail to be useful to radiologists, nuclear medicine imagers, and oncologists. 

A: thank you very much for the comments

Specific critiques

Practicing physicians can base their radiology testing on the data presented in the paper. It might be especially useful in the closing paragraphs to describe the exact imaging protocols the authors or their collaborators use. For example, MRI at question of diagnosis, PET/CT prior to start of therapy, repeat PET/CT after 3 cycles of induction therapy.

A: Yes, nice suggestion, thank you. We added in the final paragraph the sentence” Even though recommendations to the use of different techniques may vary depending on available local resources, we personally use PET/CT at diagnosis and at the end of a fixed treatment therapy (3 months after ASCT) and then every 6 months for 2 years. We integrate radiological study with MRI when PET is negative for disease and when local vertebroplasty or radiotherapy is needed”.

Along those lines, in the Conclusions, the authors bring up low expressing hexokinse-2 tumors as an explanation for false-negative PET/CT in myeloma, about 10%. How to use this knowledge prospectively? Perhaps a algorithm diagram - if pre-treatment PET/CT is negative, no use in repeating at 3 months, perhaps use MRI, BMBx in that case?

A: yes I would keep the sentence above

Line 339:  "yields false negative results losing specificity." Should be losing sensitivity, no?

A: YES correct

The English is overall outstanding but some minor edits:

Line 38: "Many progresses have been made in MM" should be "Many advances"

A: Ok thanks

Lines 257-257: "NGF with functional imaging to define response in MM, though the role of MRI to complement MRD needs confirmation." Sentence fragment. Perhaps, "NGF with functional imaging has been used to define.."

A:Ok thank you

Line 259: "Nowadays it is crucial to have a standardization of guidelines" is awkward. Maybe, "Moving forward, it is crucial to standardize guidelines?"

A: Yes thank you

Reviewer 2 Report

This is a very nice review article by Santoni etc, summarizing the use and clinical values of PET CT scan and MRI in multiple myeloma. The manuscript was well-written and included many important studies on this topic. There are a few minor comments:

Page 1, Line 34.  Suggest change “B-cell malignancy” to “plasma cell malignancy”. Myeloma is not considered to be a “B-cell malignancy”

Page 1, Line 41,  Change “³” should be “>” 1. Myeloma defining event includes more than 1 lesion.

Page 2, Line 48, “80-90%” seems high. Typically bone lesions occur in 2/3 of myeloma patients.

Line 204: ADC map needs to be spelled out. The full name of ADC was spelled in Line 232

Table 2. it would be easier for the readers to understand the meaning of the number and percentage for responders and non-responders with ADC value changes after treatment. In some studies, both number and percentage were included but in others, only percentage was included. Need to

It would be great to add a section or paragraph on the utility and value of whole body low dose CT skeletal survey.

Author Response

Reviewer.2

This is a very nice review article by Santoni etc, summarizing the use and clinical values of PET CT scan and MRI in multiple myeloma. The manuscript was well-written and included many important studies on this topic. There are a few minor comments:

 A: We thank reviewer 2 for the comments

Page 1, Line 34.  Suggest change “B-cell malignancy” to “plasma cell malignancy”. Myeloma is not considered to be a “B-cell malignancy”

A:Sure thank you

Page 1, Line 41,  Change “³” should be “>” 1. Myeloma defining event includes more than 1 lesion.

A:Yes thanks

Page 2, Line 48, “80-90%” seems high. Typically bone lesions occur in 2/3 of myeloma patients.

A: We changed in 70% of the patients

Line 204: ADC map needs to be spelled out. The full name of ADC was spelled in Line 232

A: Yes thank you

Table 2. it would be easier for the readers to understand the meaning of the number and percentage for responders and non-responders with ADC value changes after treatment. In some studies, both number and percentage were included but in others, only percentage was included. Need to

A:Yes we agree on this point , although we reported as it was showed in the papers and we are not able to change that to uniform the table.

It would be great to add a section or paragraph on the utility and value of whole body low dose CT skeletal survey.

A: Thanks again  for the consideration. Although whole body low dose CT is used in smoldering myeloma and it has been used mainly at diagnosis in  active MM in the past,  it is less used now with the availability of PET and rarely thereafter to evaluate response in active myeloma treated patients. The purpose of our manuscript was mainly to define PETCT and MRI for response evaluation. Thus, in our opinion adding that paragraph could lead a bit out of our final purpose for response evaluation.
